# Advancing 3Rs: The Mouse Estrus Detector (MED) as a Low-Stress, Painless, and Efficient Tool for Estrus Determination in Mice

**DOI:** 10.3390/ijms25179429

**Published:** 2024-08-30

**Authors:** Irina V. Belozertseva, Dmitrijs D. Merkulovs, Helena Kaiser, Timofey S. Rozhdestvensky, Boris V. Skryabin

**Affiliations:** 1Valdman Institute of Pharmacology, Pavlov First Saint Petersburg State Medical University, L’va Tolstogo str. 6-8, St. Petersburg 197022, Russia; belozertseva@gmail.com; 2ELMI Ltd., Bukultu str. 7B, LV-1005 Riga, Latvia; dmitrijs_merkulovs@yahoo.co.uk; 3Core Facility Transgenic Animal and Genetic Engineering Models (TRAM), Medical Faculty, University of Münster, von-Esmarch str. 56, D-48149 Münster, Germany; helena.kaiser@ukmuenster.de

**Keywords:** 3Rs, female mice estrous cycle, vaginal wall active resistance, mouse estrus detector

## Abstract

Determining the estrous cycle stages in mice is essential for optimizing breeding strategies, synchronizing experimental timelines, and facilitating studies in behavior, drug testing, and genetics. It is critical for reducing the production of genetically unmodified offspring in the generation and investigation of genetically modified animal models. An accurate detection of the estrus cycle is particularly relevant in the context of the 3Rs—Replacement, Reduction, and Refinement. The estrous cycle, encompassing the reproductive phases of mice, is key to refining experimental designs and addressing ethical issues related to the use of animals in research. This study presents results from two independent laboratories on the efficacy of the Mouse Estrus Detector (MED) from ELMI Ltd. (Latvia) for the accurate determination of the estrus phase. The female mice of five strains/stocks (CD1, FVB/N, C57Bl6/J, B6D2F1, and Swiss) were used. The results showed that the MEDPro^TM^ is a low-traumatic, simple, rapid, and painless method of estrus detection that supports the principles of the 3Rs. The use of the MEDPro^TM^ for estrus detection in mice caused minimal stress, enhanced mating efficiency, facilitated an increase in the number of embryos for in vitro fertilization, and allowed the production of the desired number of foster animals.

## 1. Introduction

The laboratory mouse (*Mus musculus*, L., 1758) is an important biological object for modern research in various fields of medicine and biology because of its relative similarity with the human genome and its high grade of conservation on molecular and cellular mechanisms associated with basic physiological systems (cardiovascular, nervous, endocrine, immune, etc.) [1,2,3]. The importance of the mouse as an object of biomedical research is also determined by the availability of effective technologies for the modification of the mouse genome, and the short reproductive period [4,5,6].

Identification of the estrous stages is a key step in various mouse genome bioengineering technologies and manipulations with mouse embryos. It is important to accurately stage pregnancies for evaluating prenatal pharmacological treatments and embryonic toxicity [7,8,9,10,11,12]. Additionally, determining the stages of the estrous cycle is of significant importance in diverse biomedical research applications, particularly in the context of pharmacodynamic investigations [13,14,15]. The distinct phases of the female sexual cycle exert a profound influence on the experimental process and outcomes, playing a critical role, especially in neuroscience research [16,17,18]. Efficient detection of estrus in female mice also allows efficient crossbreeding, facilitating the acquisition of pseudo-fertile (foster) mice for embryo transfer procedures [19]. This, in turn, significantly reduces the number of animals required for experiments, an important aspect of biomedical research [13,20,21]. Moreover, precise detection of the estrous cycle in mice refines experimental procedures, minimizing stress associated with unnecessary mating attempts and reducing the discomfort experienced by the animals, contributing to their well-being [19,22]. This approach ensures that experiments are conducted at appropriate times, thereby reducing variability in experimental outcomes. The resulting refinement contributes to the production of more reliable and reproducible data, enhancing the overall quality of scientific research [16,18,23,24]. Therefore, accurate determination of the estrous cycle in mice not only promotes ethical considerations in animal research but also optimizes the efficiency of experimental designs. This alignment with the 3R principles—Replacement, Reduction, and Refinement—underscores the commitment to ethical and efficient animal use in research practices [20,25].

The laboratory mouse is a polyestrous species with a reproductive cycle length of 4–5 days. An estrous cycle typically comprises four main phases, each characterized by specific hormonal changes and physiological alterations: (1) proestrus (PE), (2) estrus (E), (3) metestrus (ME), and (4) diestrus (DE), that are constantly repeated. The regulation is based on the hypothalamic–pituitary–gonadal (HPG) axis, where gonadotropin-releasing hormone (GnRH) from the hypothalamus initiates the cycle by stimulating the biosynthesis and secretion of follicle-stimulating hormone (FSH) and luteinizing hormone (LH) in the anterior lobe of the pituitary gland [26]. During proestrus, increased FSH levels activate the cAMP-protein kinase A (PKA) pathway in granulosa cells, enhancing cell proliferation and estrogen production by increasing aromatase (*Cyp19a1*) gene expression [27]. Estrogen binds to its receptors (ERα and ERβ), promoting the transcription of genes crucial for follicular development. Elevated estrogen levels trigger a LH surge through positive feedback from the hypothalamus and pituitary gland. The LH surge activates the PI3K-Akt pathway in theca and granulosa cells, leading to ovulation by promoting oocyte maturation and follicular rupture through the activation of proteolytic enzymes during the estrus stage [28]. Post-ovulation, during metestrus, the corpus luteum forms and secretes progesterone, which maintains the uterine lining (endometrium) and prepares it for implantation of a potential embryo. Key regulatory pathways include cAMP-PKA, which induces steroidogenic acute regulatory protein (*StAR*) gene activation, which mediates progesterone synthesis [29]. Progesterone binds to its receptors (PR-A and PR-B), activating transcription of genes involved in uterine maintenance and immunomodulation, such as the orphan nuclear receptor *NR4A1*, integrins, and vascular endothelial growth factor (*VEGF*) [30]. Transforming growth factor-beta (TGF-β) signaling and cytokine regulation are involved in tissue remodeling and immune regulation [31]. In diestrus, elevated progesterone levels continue to maintain the uterine environment and exert negative feedback on the HPG axis, reducing GnRH, FSH, and LH levels to prevent the development of new follicles. The interplay between these hormones and molecular pathways, including cAMP-PKA, PI3K-Akt, estrogen receptor, progesterone receptor, and TGF-β signaling, provides precise regulation of follicular growth, ovulation, and preparation for potential pregnancy [32]. Various methods for determining these estrous stages have been reported [33,34,35], including the following: (1) visual inspection of vaginal opening; (2) cytological analysis of vaginal smears/lavages; (3) quantitative measurement of sex hormones (including estrogen and progesterone) in blood plasma [36]; (4) biochemical analysis of urine and feces [37]; and (5) measurement of the impedance of the vaginal mucosa [33]. 

Visual examination of the vaginal opening, which varies from maximally dilated during estrus to constricted during diestrus, is a straightforward but subjective and severely limited method. Notably, significant variability exists between various strains of laboratory mice concerning the appearance of the vagina during different estrous cycle stages [34,38]. The analysis of sex hormones in blood plasma and the biochemical examination of urine and feces provide a more precise determination of the estrous cycle stage. However, these methods are time-consuming and require the presence of qualified personnel and expensive equipment. In addition, daily blood sampling for biochemical analysis results in a reduction in circulating blood volume, which is painful and stressful for animals. Therefore, it is not suitable for mass screening and monitoring of animals. The accuracy of detection of the sex hormones in urine and feces is limited and strongly depends on the particular mouse strain employed in the experiment [39]. Cytological examination of the vaginal smear to identify the predominant cell types present during different stages of the estrous cycle is the most accurate method, and it is accepted as the “gold standard” for determining distinct stages of the reproduction cycle in rodents [35,40,41]. Despite its precision, this method is time-consuming and requires the expertise of qualified personnel. Consequently, it is also not suitable for large-scale animal screening and monitoring. Measurement of the impedance of the epithelial cell layer of the vaginal mucosa to determine the estrous cycle stages in rodents is the simplest, most cost-effective method for obtaining data on ovulation [42,43]. Historically, the development of detectors to determine ovulation time by assessing the impedance of the vaginal walls in farm animals traces back to 1961. During this period, the USSR issued a certificate of authorship №736350/30-1 for the “Method for determining the optimal time of inseminating cows”, followed by the reception of patent USSR No. 178602 for a device invented for measuring the electrical resistance of the vaginal walls of cows. The application of vaginal impedance measurements was introduced for the first time in laboratory animals in 1977, specifically rats [44] and guinea pigs [45]. Commercial devices engineered to measure impedance in laboratory rodents were subsequently developed by Muromachi Kikai Co., Ltd. (Chūō, Japan) and designated as rat vaginal impedance checkers MK-11 and MK-12. These devices proved remarkable effectiveness in determining the estrus in rats. However, the measurement applicability in mice has been neither guaranteed nor verified by the manufacturer (muromachi.com).

In this study, we present an integrated and independent comprehensive analysis of a novel Mouse Estrus Detector (MED) MEDPro^TM^ device developed by ELMI Ltd. (Riga, Latvia) [46], which is specifically designed for the quantification of vaginal mucosa active resistance in mice.

## 2. Results

### 2.1. Comparison of Vaginal Smear Cytology, Vaginal Opening, and AR Value

To evaluate the effectiveness of the MEDPro^TM^ and determine an active resistance (AR) value for detecting estrus stage in mouse cycles, we conducted a comprehensive series of experiments involving CD-1, FVB/N, and C57Bl6/J female mice (n = 10 for each strain). We simultaneously performed observational assessments of vaginal opening, AR measurements, and vaginal smear sample cytology on the same animals over an 11-day period, with daily evaluations for each mouse in the respective order (Appendix A).

We collected a total of 330 vaginal smear samples from females (11 vaginal smears per animal) of the investigated strains. Notably, each strain demonstrated different estrous cycle regularity, with variations observed in the continuous progression from proestrus to estrus to metestrus and diestrus stages. We observed a cycle regularity of approximately 60% for CD-1, 80% for FVB/N, and 40% for C57Bl6/J strains.

Comparison of vaginal opening assessment with vaginal smear analysis revealed pronounced differences depending on the mouse strain examined (Figure 1). While vaginal opening observations were made at each estrous stage of the investigated strains, the percentages of estrus verification varied. Specifically, vaginal opening correlated with estrus determination in 63% of cases for the CD-1 strain and 39% for the FVB/N and C57Bl6/J strains (Appendix A).

Conversely, the vaginal opening was detected in 33% of metestrus and 26% of diestrus cycles in CD-1 and in 35% of metestrus and 8% of diestrus cycles in the FVB/N strain (Appendix A). In contrast, C57Bl/6J mice exhibited vaginal opening in only 8% of metestrus cycles and 0% of diestrus cycles (Appendix A). Based on the cytological determination (used as the gold standard) of the estrus stage in the mouse strains tested, we compared the proportions of mice in estrus among females with open and closed vaginas (Figure 1) using the Chi-square test. The comparison showed a significant difference in these proportions for CD-1 (χ^2^ = 6.133, df = 1, *p* = 0.013) and C57Bl6/J (χ^2^ = 8.067, df = 1, *p* = 0.005), but not for FVB/N (χ^2^ = 1.611, df = 1, *p* = 0.204). Thus, this method was ineffective in determining estrus in FVB/N females.

The obtained results highlight the high variability in the efficacy of vaginal opening observation across mouse strains and underscore the importance of utilizing complementary methods for accurate estrus determination. 

Following the assessment of vaginal opening and prior to vaginal smear sampling, we performed daily measurements of vaginal AR in the investigated mice using the MEDPro^TM^ device (Figure 2 and Figure 3, Appendix A). A two-way mixed model analysis of variance (ANOVA) was conducted to analyze the data. The results revealed significant differences between estrous cycle stages in the value of AR for each strain: CD-1 (F(3,44) = 13.53; *p* = 0.000); FVB/N (F(3,39) = 36.62; *p* = 0.000); and C57Bl6/J (F(3,50) = 48.22; *p* = 0.000). In female mice of all strains, AR was significantly higher during estrus compared to other stages of the cycle (*p* < 0.001). Simultaneously, the AR of the vaginal wall was at its lowest values in diestrus (Figure 4 and Appendix A, and Appendix A).

Based on the determination of vaginal wall AR in females from investigated mouse strains, an AR value of 6 kOm was chosen as the threshold for identifying estrus. The actual probability of detecting the estrus in the next round of measurements, given AR values equal to or greater than 6 kOm, was calculated as follows: 86% for CD-1, 64% for FVB/N, and 76% for C57Bl/6J mouse strains. Comparison of the proportion of mice in estrus among females with AR < 6 kOm and AR > 6 kOm (Figure 5) showed highly significant differences in all strains studied: CD-1 (χ^2^ = 85.924, df = 1, *p* < 0.001), FVB/N (χ^2^ = 53.539, df = 1, *p* < 0.001), and C57Bl6/J (χ^2^ = 64.744, df = 1, *p* < 0.001). 

However, given the substantial interindividual (interstrain) fluctuations observed in vaginal AR (Appendix A), it is crucial to emphasize that the determination of the AR threshold for each particular mouse strain or hybrid should be established experimentally before integrating MedPro^TM^ into laboratory protocols.

### 2.2. Estimation of Mating Efficiency for Female Mice with Different Vaginal AR Values

The reliable correlation between the MEDPro^TM^ measurements and vaginal smear cytological results in detecting the estrus stage prompted us to test the device for refining mating efficiency. We assessed mating efficiency by examining the presence of the copulatory plug after overnight pairing of CD-1 male mice with females characterized by different values of vaginal AR determined with MEDPro^TM^. As anticipated, the highest presence of the copulatory plug (60%) was detected after overnight pairing of CD-1 male mice with females with AR values exceeding 6 kOm (*χ*^2^ = 17.620, *df* = 1, *p* < 0.001) (Figure 6). This corresponds with the estrus and, putatively, partly with the proestrus stages, which align with the ovulation phase. Notably, the vaginal plug in mice can sometimes be small and lie deeply in the vagina, or it can dissolve and become not visible [47]. Therefore, it is still possible that mating occurred in a higher number of animals than observed.

Hence, we additionally evaluated the significance of the MEDPro^TM^ application for successful timed pregnancies, using male and female Swiss mice. The proportion of Swiss females that gave birth (Figure 6) was significantly higher (*p* = 0.02, Fisher’s exact test) in the group of females with detected AR > 6 kOm (6 out of 9) compared to the “Undetected females” group (1 out of 10).

### 2.3. MEDPro^TM^ Application for Determining Female Mice Superovulation

In our attempt to optimize the superovulation protocol, we investigate the impact of the estrous stages on superovulation in mature female mice. The method of mouse superovulation, initially described in 1956 [39], has been employed in transgenesis since the 1980s, enabling the generation of a substantial number of oocytes from a limited number of female mice [48]. While natural ovulation typically yields approximately 6–10 oocytes per female, superovulation can significantly increase the number, often reaching up to 60 oocytes [49]. This process involves the intraperitoneal injection of gonadotropin hormones, which mimic natural mouse hormones and initiate the maturation of numerous oocyte follicles. It is noteworthy that superovulation, akin to normal ovulation, triggers female receptivity and enhances its attraction for males. In this study, we utilized the MEDPro^TM^ device to investigate two inbred FVB/N and C57Bl6/J strains and one hybrid strain, B6D2F1 (C57Bl6/J × DBA). Similar to previous experiments, an AR value exceeding 6 kOm was used as the threshold to determine the estrus stage in mice. Our results showed that the intraperitoneal injections of gonadotropin hormones during the estrus stage (AR > 6 kOm) increased the overall number of obtained oocytes. Remarkably, while accurate determination of the estrous cycle with MEDPro^TM^ had a less pronounced impact on the FVB/N and C57Bl6/J inbred strains (17% and 33%, respectively), it resulted in an over 50% increase in the number of superovulated oocytes in the B6D2F1 hybrid strain (Figure 7). Notably, the generation of most genetically modified mice in our laboratory is performed by microinjecting CRISPR-Cas9 components into in vitro fertilized one-cell oocytes (zygotes) of the B6D2F1 hybrid strain, which shows high survival rates compared with commonly used inbred strains [50,51].

### 2.4. Assessment of Distress Caused by Measuring the AR with MEDPro^TM^


Locomotor activity data were recorded (Appendix A) and analyzed, confirming a significant influence of the factor “Experimental group” on general horizontal motor activity (H = 9.153; df = 2; *p* = 0.010) and ambulation (H = 7.687; df = 2; *p* = 0.021). The influence on vertical activity (H = 5.823; df = 2; *p* = 0.054) and the number of fecal boli (F (2,53) = 2.759; *p* = 0.072) was close to significant.

Female mice, 30 min after vaginal lavage, exhibited a statistically significant decrease in horizontal activity, including ambulation, compared with the intact control group (Figure 8). The decrease in vertical activity and fecal boli count (indices of emotionality and anxiety) [52] in this group was close to significant. In contrast, animals did not exhibit a statistically significant reduction in either horizontal activity, including ambulation, or in vertical activity and fecal boli counts after AR measurement compared to the intact control group (Figure 8).

## 3. Discussion

Understanding the reproductive cycles of animals is essential in various scientific fields, from basic research to toxicological and pharmacological studies. In this context, the accurate determination of estrus, a key phase in the reproductive cycle of female mammals, holds particular significance. Traditional techniques for estrus detection, such as vaginal lavage, can induce stress in animals, potentially biasing experimental results [24,53].

While adopting a new experimental technique, a thorough evaluation of its impact on animal wellbeing is essential. Assessment of stress levels introduced by the technique is important not only for animal wellbeing and ethical consideration but also for ensuring accurate experimental result interpretation and drawing valid conclusions [54]. The measurement of locomotor activity often serves as an indicator of stress and an assessment of the basic status of animals. The decline in locomotor activity in mice may correspond to procedural distress [55]. There has been accumulating evidence showing that a variety of stimuli affect the motor activity of laboratory rodents [52,55,56,57,58]. Such mild stress as grabbing the mice by the tail and neck skin (five times) before placing them in the open field significantly reduced the average travel distance [58]. The results of our study revealed that 30 min after vaginal lavage, female Swiss mice exhibited a statistically significant (but not dramatical) decrease in horizontal activity, including ambulation, and a close to significant decline in vertical activity, along with an increase in defecation level, indicative of heightened emotionality under low levels of novel environmental stress conditions (Figure 8) [52]. In contrast, this pattern was not observed following the insertion of the MEDPro^TM^ detector probe and measurement of estrous cycle-related vaginal AR, indicating that this method is a less stressful and more refined approach for estrous stage evaluation.

To assess the efficacy of the MEDPro™ device in detecting estrus stage, we conducted a comparative study across CD-1, FVB/N, and C57Bl6/J mouse strains. We designed a comprehensive series of experiments spanning an 11-day period, wherein we performed daily observational assessments of vaginal opening, measured active resistance (AR), and compared the results with vaginal smear cytology (Figure 9, Appendix A). Our study collected a total of 330 vaginal smear samples, revealing different estrous cycle regularities among the investigated strains. When a comparison of vaginal opening with vaginal smear analysis was performed, the rates of estrus confirmation varied significantly, underscoring the variability in the effectiveness of vaginal opening observation across mouse strains and emphasizing the necessity of complementary methodologies for precise estrus determination [34]. To assess differences in vaginal active resistance at different stages of the estrous cycle obtained with the MEDPro^TM^ device, we used mixed model rank analysis for each strain of mice. Our results demonstrated a significant (*p* < 0.01) difference in AR values between stages in all strains (Figure 4, Appendix A). Specifically, AR was significantly higher during estrus than during other cycle stages, with the lowest values observed during diestrus. In assessing the likelihood of detecting each estrus cycle in the subsequent round of measurements based on AR values, our data revealed that values equal to or greater than the threshold of 6 kOm resulted in detection probabilities of approximately 76%, 64%, and 76% for CD-1, FVB/N, and C57Bl6/j strains, respectively. Conversely, the probabilities for detecting proestrus and metestrus were comparatively lower. Importantly, values equal to or greater than 6 kOm were not detected for diestrus cycles in any of the investigated mouse strains.

The estrous cycle in mice is regulated by a complex interplay of hormonal and signaling pathways, primarily coordinated by the hypothalamic-pituitary-gonadal (HPG) axis [28,59,60]. This axis functions through the release of gonadotropin-releasing hormone (GnRH) from the hypothalamus, which stimulates the anterior pituitary to secrete follicle-stimulating hormone (FSH) and luteinizing hormone (LH). FSH is pivotal for follicular growth, cellular proliferation, and estrogen production, whereas LH is essential for androgen biosynthesis, oocyte maturation, ovulation, and the differentiation of follicles into the corpus luteum (CL) [28,59,60]. During the proestrus phase, increased FSH levels enhance follicular development and estrogen production. High estrogen levels activate an LH surge that triggers ovulation, marking the transition to the estrus phase and promoting oocyte maturation and follicular rupture. Following ovulation, in the metestrus phase, the corpus luteum forms and secretes progesterone, maintaining the uterine lining for potential embryo implantation. Progesterone remains elevated during diestrus, sustaining the uterine environment and exerting negative feedback on the HPG axis to prevent the development of new follicles. This hormonal interplay, involving FSH, LH, estrogen, and progesterone, precisely regulates follicular growth, ovulation, and preparation for potential pregnancy. Mouse mating typically aligns with the estrus phase, where conditions are optimal for fertilization. Accurate timing of estrus is critical for successful mating and superovulation, which are essential in various scientific disciplines, including reproductive biology, developmental biology, embryology, toxicology, pharmacology, genetics, and biotechnology. These fields often require precise control over reproductive cycles to study processes such as fertilization, embryonic development, the effects of drugs or chemicals on reproduction, and the generation of genetically modified organisms. Traditionally, timed pregnancies are determined by housing male and female mice together overnight and checking for mating signs (“plug”) the following morning. Our findings demonstrate that integrating MEDPro^TM^ into the natural breeding protocol significantly enhances mating efficiency by predominantly selecting female mice in the estrus stage of the cycle, offering a streamlined approach to research practices in accordance with the 3Rs principles to refine animal mating (Figure 6) [20,25,54].

The protocol for superovulation in mice is based on the injection of pregnant mare serum gonadotropin (PMSG), a hormone that has the activity of both FSH and LH hormones, followed by an injection of human chorionic gonadotropin (hCG), which has LH-like activity. Hence, the PMSG enhances follicle development and promotes ovulation and luteum formation, whereas the hCG stimulates final follicle maturation and ovulation. This technique aligns with the principles of the 3Rs in animal research, drastically minimizing the number of animals utilized in experiments and enabling the generation of substantial numbers of embryos for various applications [20,61]. In vitro fertilization (IVF), embryo transfer, and embryo cryopreservation are just a few examples of reproductive procedures that rely on superovulated oocytes. Superovulation stands as a crucial step for the generation of genetically modified mouse models through microinjections of zygotes and/or embryonic stem cell delivery into blastocytes. However, it has also been described that different strains of mice and animal weight can potentially influence hormone levels and, therefore, the efficiency of superovulation [62]. Moreover, it was reported that gonadotropin administration in mice should be synchronized with the natural estrous cycle to optimize oocyte quality [63]. Injecting PMSG at diestrus results in a higher percentage of cumulus-free oocytes, oocytes without a polar body, oocytes with intracytoplasmic mitochondrial aggregates, and those with abnormal chromosome distribution and scattering. In contrast, estrus females show the highest percentage of oocytes with normal chromosome distribution and the lowest percentage of oocytes with the aforementioned abnormalities [63]. In addition, the highest pregnancy rates were observed in mice that superovulated during the proestrus and estrus stages [64]. The use of suboptimal mouse strains and/or suboptimal superovulation protocols results in the use of a large number of females to obtain the necessary number of oocytes [61,65,66,67,68]. Given the considerable genetic variations between mouse strains, establishing an optimal superovulation protocol becomes crucial [69]. In this study, we aimed to enhance the superovulation protocol for FVB/N, C57Bl6/J, and B6D2F1 mouse strains that are routinely used for genome editing in our laboratory. Notably, embryonal survival after CRISPR/Cas9 complex microinjection and efficient implantation after embryo-transfer procedures are among the crucial aspects of the generation of genetically modified animals that might directly correlate with the quality of the obtained oocytes after superovulation (Skryabin et al., unpublished). In our analyses, prior to hormonal injections, mice were separated into two groups based on the value of vaginal active resistance. Although, due to technical and ethical limitations, counting the number of oocytes from individual animals was not feasible, our findings suggest that initiating superovulation during the estrus cycle (AR ≥ 6 kOm) consistently resulted in higher oocyte yields across all females, irrespective of mouse strain. Notably, certain strains, such as FVB/N and C57Bl6/J, exhibited slightly enhanced embryo production, while the B6D2F1 hybrid strain displayed a remarkable increase of up to 50 percent more embryos per female animal (Figure 7). This discovery is particularly significant, as the high survival rates of B6D2F1 zygotes make them the primary choice for CRISPR-Cas9 mediated genome editing in our research [50,51]. In summary, using the MEDPro^TM^ device allows for a fast, efficient, and minimal impact on the wellness of female mice to determine their estrus cycle. Compared to the other non-traumatic visual method, the MEDPro^TM^ device allows for a more accurate detection of the estrus cycle. 

## 4. Materials and Methods 

### 4.1. Animals

In this study, a number of experiments were carried out in female mice of different strains (CD-1, FVB/N, and C57Bl6/J) and hybrid strains, B6D2F1 (C57Bl6/J × DBA); see the detailed description below. Thirty pubertal female mice (ten for each line: CD-1, FVB/N, and C57Bl6/J) were assessed daily for stage of the estrous cycle by assessing vaginal opening status, MEDPro^TM^ measurements, and vaginal smear cytology. A two-way mixed model analysis of variance (ANOVA) was performed on the active resistance data after MEDPro^TM^ measurements. Animals were kept in specific pathogen-free conditions at the transgenic animals and genetic engineering models (TRAM) core facility of the University Clinic Muenster. All female mice were housed in groups (five animals per cage, with non-coniferous wood-chip bedding, Wilhelm Reckhorn GmbH, Warendorf, Germany) with free access to filtered tap water and food (Altromin, 1324, Lage, Germany). They were maintained under a 12 h light/dark cycle (lights on at 06:00 AM) at an ambient temperature of 20 ± 1 °C and humidity of 50 ± 10%. All mouse procedures with the above-mentioned animals were performed in compliance with the guidelines for the welfare of experimental animals issued by the Federal Government of Germany and approved by the State Agency for Nature, Environment and Consumer Protection North Rhine-Westphalia LANUV (Landesamt für Natur, Umwelt und Verbraucherschutz Nordrhein-Westfalen).

In addition, the female (n = 75) and male (n = 19) outbred Swiss mice from a local colony of the Department of Psychopharmacology (Valdman Institute of Pharmacology, St. Petersburg, Russia) were used for behavioral tests and estimation of mating efficiency. Animals were housed in groups of same-sex siblings (three to five mice per TIII cage, Velaz, Czech Republic; with non-coniferous wood-chip bedding, Lignocel, JRS GMBH + Co., Pattensen, Germany) with free access to filtered tap water (filter AQUAPHOR^®^, St. Petersburg, Russia) and food (standard lab chow, Laboratorkorm, Moscow, Russia). They were maintained under a 12 h light/dark cycle (lights on at 09:00 a.m.) at an ambient temperature of 20 ± 1 °C and humidity of 50 ± 10%. Cages, bedding, and water bottles were changed at regular intervals, i.e., twice a week.

### 4.2. Vaginal Opening Observation 

Vaginal opening observation is a commonly used technique due to its non-invasive nature and simplicity. However, it exhibits significant variability across mouse strains and demonstrates low efficacy in accurately determining estrus stages [34,40,70]. During estrus, the vaginal opening typically appears open, swollen, protruding, and moist with a reddish or pinkish hue, whereas during proestrus, it remains relatively closed and less protruded. Metestrus represents the transitional phase between estrus and diestrus, characterized by a decrease in swelling and moisture and a return to a more closed and constricted appearance. The reddish or pinkish coloration diminishes during this phase, and during diestrus, the vaginal opening appears minimally smaller, dry, and pale.

### 4.3. General Principles of MEDPro^TM^ Design and Measurement 

Measuring the impedance of the vaginal walls in mice poses challenges in accurately determining the stage of the estrous cycle. In existing rodent estrous cycle detectors, the electrical resistance of the vaginal mucosa is determined by measuring the electrical impedance (Z), which is the geometric sum of the active resistance of the R_S_ electric circuit and the reactive resistance of the X_C_: Z^2^ = R_S_^2^ + X_C_^2^. R_S_ is an inverse value of tissue conductivity, exhibiting minimal dependence on current frequency, while the reactive component of the X_C_ impedance depends on the frequency of the electric current. In alternating current measurements, Z is influenced by system interfaces in the system, where charge accumulation—polarization—can occur. The properties of interfaces (in a biological object, these are mainly different cell membranes) can be described by taking into account the capacitance C, the resistance of which X_C_ depends on the frequency at which the measurement is made: X_C_ = −1/(ωC), where ω = 2πf; ω—circular frequency; f—frequency in Hz. Measurement of conductivity in biological systems with direct current is challenging due to significant polarization on the surfaces of cell membranes and probe electrodes. Therefore, alternating current is employed for measurements.

In 2015, ELMI Ltd. (Riga, Latvia) initiated the development of the MEDPro^TM^ device, primarily focused on determining the vaginal AR in female mice (patent LV15278) [46]. Several versions of the device have been developed and tested, preceding the one presented in this work [42]. These units (Table 1) employ the synchronous detection method, selectively measuring the active resistance of the *R_S_* components, thereby eliminating the influence of C as “parasitic” noise on the measurement result. In contrast to the larger probes of MK-11 and MK-12 (Muromachi Kikai Co., Ltd., Japan) estrous cycle detectors, which require time for temperature stabilization of the index, the MEDPro^TM^ units feature a miniature probe (Figure 2). This probe comprises two electrodes (∅ = 1.82 mm, 1.2 mm wide, AISI316 material, 1.2 mm electrode spacing) with a low heat capacity, facilitating rapid (<1 s) temperature equalization between probe and tissue. Moreover, the miniature probe (L20xO.D.1.82), when inserted into the mouse vagina (Figure 3), is less traumatic and induces less stress, contributing to improved measurement accuracy.

The device operates in two modes: “Auto” and “Manual” (Figure 2). In “Manual” mode, the sinusoidal voltage can be adjusted manually. In “Auto” mode, the device automatically regulates the current passing through the sensor electrodes of the probe to ensure it does not exceed 1 µA. If the current approaches this threshold during measurement, the device automatically reduces the sinusoidal voltage in real-time. This regulation is necessary because, at low voltages and high resistance, the resulting current is extremely small and noisy, complicating the accurate determination of impedance components *R_S_* and *X_C_*. All measurements reported in this study were conducted in “Auto” mode.

### 4.4. Vaginal Smear Cytology for Mouse Estrous Cycle Stage Identification 

Estrus cycle monitoring and stage classification were performed using vaginal cytology. A vaginal smear offers a reliable method (“gold standard”) for determining each estrous cycle stage [34,40,41]. A lavage from each animal was then taken after AR detection. Mouse vaginal smears were collected daily for 11 consecutive days in the morning (11:00 a.m.) and were used as a control assessment for comparison with other methods tested (Appendix A). The vagina was washed by pipette with 20 µL of 0,9% NaCl to suspend the epithelial and immune cells. The resulting fluid was placed on a slide and allowed to dry. The cells were stained with 0.1% crystal violet for 1 min, washed with ddH_2_O, dried, evaluated microscopically, and photographed (Figure 9) [40].

The cytology of mouse estrous cycle stages is characterized by regularly repeating combinations of distinct cell types: leucocytes—L, nuclear epithelial cells—N, and cornified epithelial cells—C (Figure 9A–D). The presence of these cell types correlates with the condition of the vaginal mucosa, is related to the state of the ovaries and uterus, and depends on the levels of different sex hormones circulating in the blood. In the standard mouse estrous cycle (4–5 days), the first stage, proestrus, is characterized by abundant nucleated epithelial cells (Figure 9A) and lasts approximately one day, followed by estrus. The estrus is usually identified by large numbers of cornified (keratinized) epithelial cells with irregular margins (Figure 9B). The predominance of keratinized cells in the vagina lasts for one to two days (depending on the 4 or 5 days of the estrus cycle) and continues until the metestrus stage, which occurs after ovulation. Metestrus is a transitional period between estrus and diestrus, characterized by leukocytes, nucleated epithelial cells, and significant numbers of cornified epithelial cells (Figure 9C). The metestrus lasts about one day and is followed by the diestrus stage, where leukocytes are present together with rounded epithelial cells with nuclei in varying concentrations. The vaginal smear can often appear almost exclusively leucocytic during diestrus (Figure 9D).

### 4.5. Mouse Mating

We assessed mating efficiency by examining the presence of the copulatory plug after overnight pairing (at a 1:1 ratio) of vasectomized CD-1 male mice with females characterized by different values of vaginal AR determined with MEDPro^TM^. Male mice were vasectomized at approximately 12 weeks of age and subsequently housed individually. To evaluate the significance of the MEDPro^TM^ application for successful timed pregnancies, sexually experienced male and sexually naive female Swiss mice (4 months old) intended for breeding were used. The study included two experimental groups. The first group of animals (“Undetected females”) was paired with males at a ratio of 1:1 in a TII cage (Tecniplast, Italy) in the afternoon and left overnight without prior estimation of the AR (n = 10). The second group (“Detected females”) consisted of females that were paired with males under the same conditions, but only after detecting an AR > 6 kOm (n = 9). In both groups, the pairing was conducted under the same environmental conditions to ensure consistency.

### 4.6. Female Mice Superovulation 

Superovulation in female mice was induced using two hormones: pregnant mare’s serum gonadotrophin (PMSG), which has follicle-stimulating activity, and human chorionic gonadotrophin (hCG). Both hormones were administered intraperitoneally. PMSG (7.5 U) was injected at 18:00, followed by an injection of hCG (7.5 U) 44–48 h later. The next day, unfertilized oocytes were isolated from the superovulated female mice.

### 4.7. Behavioral Test

Female Swiss mice (n = 56) were used in this study. The mice were divided into three groups: the MEDPro^TM^ group (n = 21), the vaginal lavage group (n = 14), and the intact control group (n = 21). Locomotor activity was assessed 30 min after the AR measurement or vaginal lavage. Tests were conducted between 14:00 and 18:00 h in Plexiglas boxes (internal dimensions: 35 × 24 × 33 cm) with transparent walls and an opaque floor. These boxes were housed in sound- and light-attenuating ventilated cubicles with dim illumination and a fan for masking noise, classified as “low-stress environments”. Locomotor activity was measured using three pairs of photobeams placed 2 cm above the floor. Consecutive crossings of different photobeams were counted as ambulation, while crossings of any photobeam were calculated as general horizontal activity. Additionally, rearing behavior (vertical activity) was scored using eight pairs of photobeams placed 10 cm above the floor. Behavioral measures were collected and recorded using IBM-compatible computers equipped with specialized hardware and software from Med Associates, Inc. (East Fairfield, VT, USA) (Appendix A). Mice were placed in the chambers for a 30-min duration. After each session, fecal boli were counted and removed. The testing surface was cleaned with 3% hydrogen peroxide and dried with paper towels before the next test. 

### 4.8. Statistical Analysis

Statistical analyses were performed using SPSS for Windows (ver. 22, SPSS Inc., Chicago, IL, USA) and SigmaPlot (ver. 12.5; Systat Software Inc., Chicago, IL, USA). The data were checked for normality of distribution using the Shapiro–Wilk test. Where necessary, due to conditions of non-normality, the data were transformed prior to analysis. A two-way mixed-model analysis of variance (ANOVA) was performed on the active resistance data. Degrees of freedom are presented rounded down to the nearest integer. Bonferroni’s test was applied for post hoc comparisons whenever indicated by the ANOVA results. Categorical data were analyzed using the Chi-square or Fisher’s exact test and reported as frequency (percentage). The data for locomotor activity were subjected to a Kruskal–Wallis one-way analysis of variance by rank test. Dunn’s tests were applied for post hoc comparisons whenever indicated by the Kruskal–Wallis analysis results. The level of significance was set at 0.05.

## 5. Conclusions

Our results emphasize the importance of employing advanced techniques, such as the MEDPro^TM^ device, in animal research, especially in advancing the principles of refinement, reduction, and replacement (3Rs). By reducing stress-induced bias in experimental results and optimizing reproductive procedures such as superovulation and animal mating, MEDPro^TM^ improves research practices while reducing animal requirements. Furthermore, our study demonstrated its efficacy in determining estrus stages in different strains of mice, highlighting the possibility of its use as a complementary technique for accurate estrus detection. We believe that the integration of MEDPro^TM^ into research protocols has the potential to improve the wellbeing of laboratory animals and enhance the reliability and ethical integrity of scientific research.

## Figures and Tables

**Figure 1 ijms-25-09429-f001:**
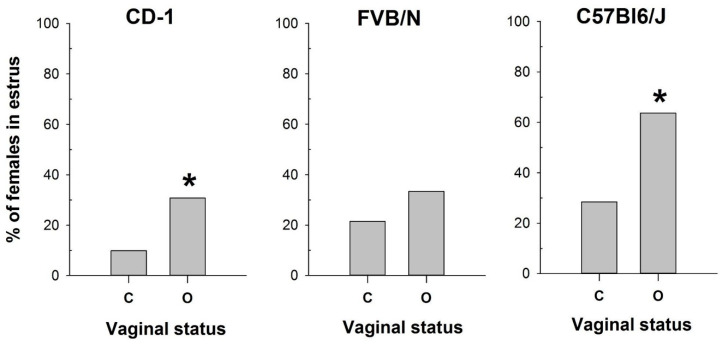
The proportion of female mice with a cytologically estimated estrus with a closed (C) or open (O) vagina: CD-1 (n = 71 and n = 39 for C and O, respectively); FVB/N (n = 79 and n = 31 for C and O, respectively); C57Bl6/J (n = 88 and n = 22 for C and O, respectively). *—*p* < 0.05 (Chi-square test).

**Figure 2 ijms-25-09429-f002:**
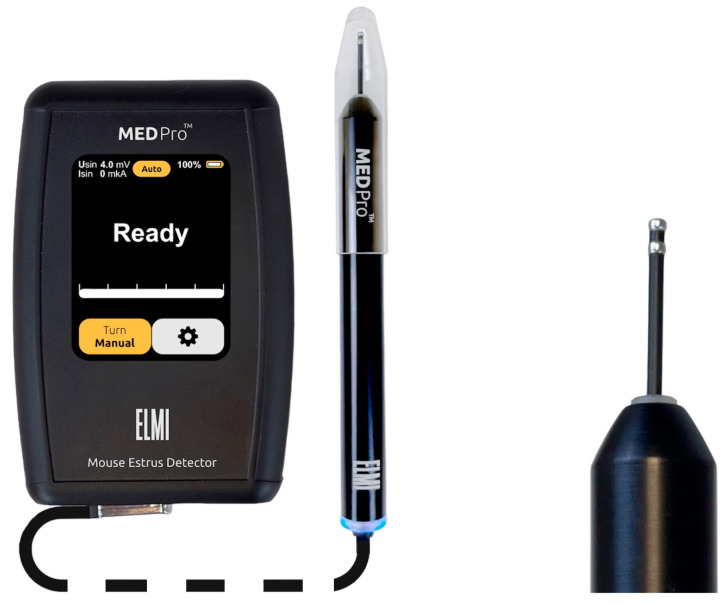
Photo of the mouse estrus detector MEDPro^TM^ device (**left panel**) and a magnified image of its probe (**right panel**).

**Figure 3 ijms-25-09429-f003:**
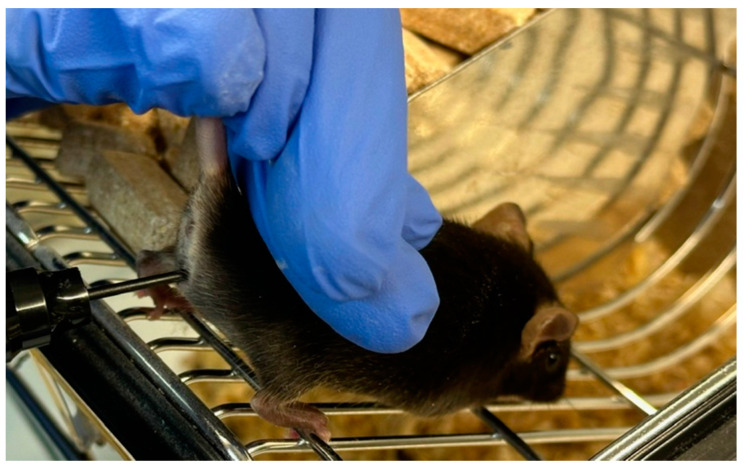
Measurement of vaginal wall active resistance in female mice by using the MEDPro^TM^ device.

**Figure 4 ijms-25-09429-f004:**
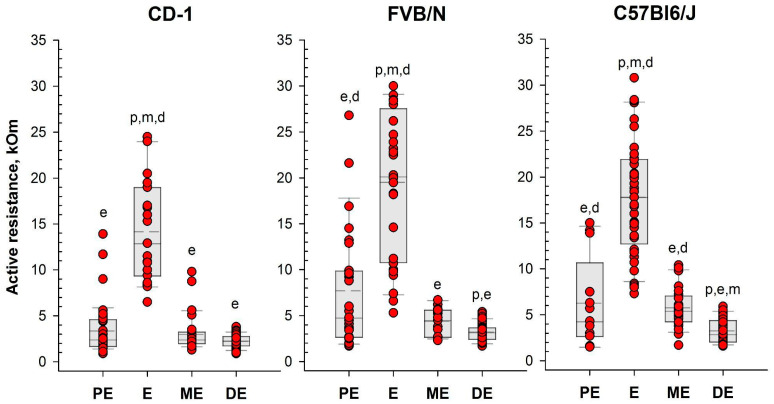
Vaginal active resistance values as a function of the stage of the estrous cycle (PE—proestrus, E—estrus, ME—metestrus, DE—diestrus), determined by vaginal smear in female mice of different strains. Single-value plots show each observation’s value. The box represents the 25th and 75th percentiles and illustrates the median (*solid line*) and the mean (*dashed line*). The whiskers show the 10th and 90th percentiles. Letters indicate a significant difference (*p* < 0.05, Bonferroni test for multiple comparisons) as follows: p—proestrus, e—estrus, m—metestrus, d—diestrus.

**Figure 5 ijms-25-09429-f005:**
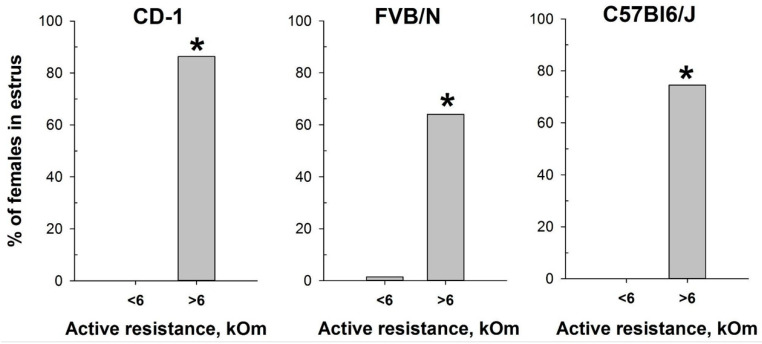
Proportion of female mice with cytologically estimated estrus with AR < 6 kOm or AR > 6 kOm: CD-1 (n = 88 and n = 22, respectively); FVB/N (n = 68 and n = 42, respectively); C57Bl6/J (n = 55 and n = 55). *—*p* < 0.001 (Chi-square test).

**Figure 6 ijms-25-09429-f006:**
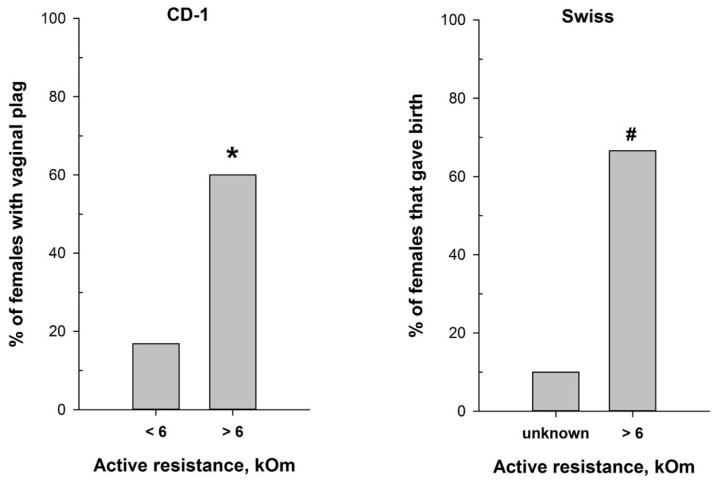
Mating efficiency in female mice with different values of vaginal active resistance after overnight pairing with male. (**left panel**)—proportion of CD-1 females with vaginal plug (n = 97 for AR < 6 kOm; n = 25 for AR > 6 kOm); (**right panel**)—proportion of Swiss females that gave birth (n = 10 for ‘unknown”; n = 9 for AR > 6 kOm). *—*p* < 0.001 (Chi-square test); #—*p* < 0.001 (Fisher’s exact test).

**Figure 7 ijms-25-09429-f007:**
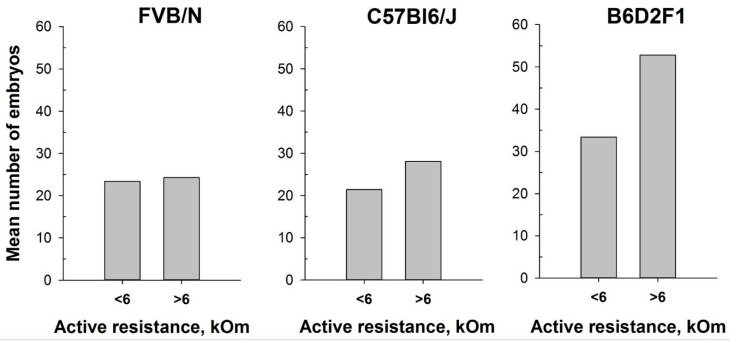
Number of oocytes per mouse produced after mouse superovulation started at different vaginal AR values—AR < 6 kOm or AR > 6 kOm: FVB/N (n = 19 and n = 5, respectively); C57Bl6/J (n = 29 and n = 15, respectively); B6D2F1 (n = 51 and n = 9, respectively).

**Figure 8 ijms-25-09429-f008:**
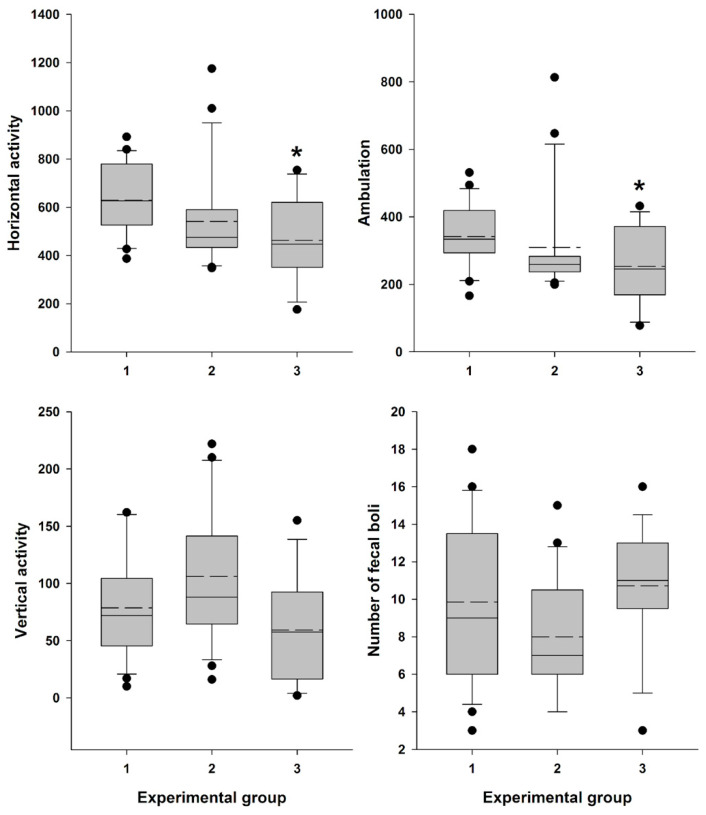
Locomotor activity and “emotionality” of Swiss female mice. A box-plot and whisker graph show the difference between the three experimental groups: (1) intact female mice (n = 21); (2) female mice 30 min after detector probe insertion into the vagina for measurement of active resistance (n = 21), and (3) female mice 30 min after vaginal lavage (n = 14). The box represents the 25th and 75th percentiles, and illustrates the median (*solid line*) and the mean (*dashed line*). The whiskers show the 10th and 90th percentiles and outliers are indicated as dots. * Indicate a significant difference (*p* < 0.05, Dunn’s test for multiple comparisons) from intact females.

**Figure 9 ijms-25-09429-f009:**
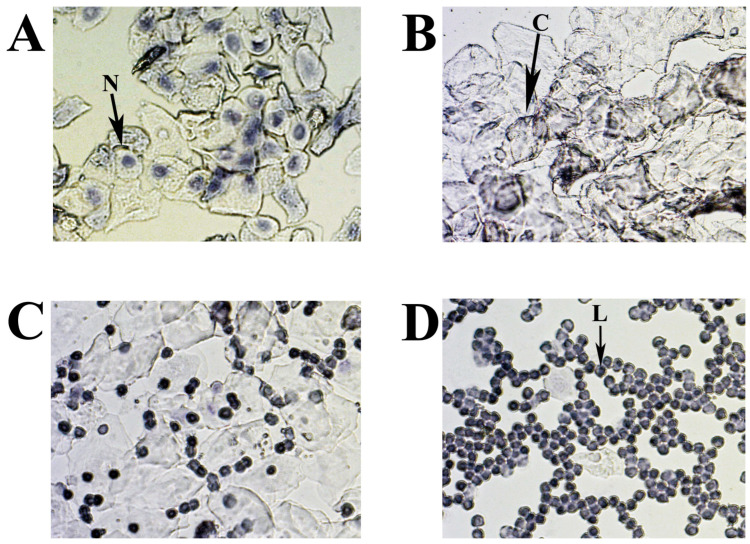
Representative smears illustrating cytology for each estrous cycle stage: (**A**) proestrus; (**B**) estrus; (**C**) metestrus; (**D**) diestrus. Three cell types are identified: leukocytes (L), cornified epithelial cells (C), and nucleated epithelial cells (N).

**Table 1 ijms-25-09429-t001:** The specifications of the MedPro^TM^ (ELMI Ltd., Latvia).

Measurement Range, kΩ	0–50
Accuracy, kΩ	0.1
Operating frequency, kHz (sinusoidal)	1
Mode	Auto, Manual
AV (sinusoidal) through probe electrodes (Auto and Manual mode), mV	0.4–4.0
Maximum AC through probe electrodes (Auto mode), µA	<1
Maximum DC through probe electrodes (Auto and Manual mode), µA	<1
Probe, mm	L20xO.D.1.82
Dimensions (without cable), mm	L138xO.D.12.5
Weight (without cable), g	30
Color graphic 2.4” Touch Display, Dimensions, mm	117 × 79 × 33
Weight (including batteries), g	220
Battery, 1.5V, AA, element	3
Battery Life (when used continuously), hour	>12
Cable AUX Jack 3.5mm Male to Male Audio for Headphones, m	1

## Data Availability

The data presented in this study are available on request from the corresponding author.

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
