# Peer review of "Advancing 3Rs: The Mouse Estrus Detector (MED) as a Low-Stress, Painless, and Efficient Tool for Estrus Determination in Mice"

_ijms, 2024, doi:10.3390/ijms25179429_

Round 1

Reviewer 1 Report

Comments and Suggestions for Authors

This manuscript presents an examination of a new tool to detect estrus in mice. The rationale is that current methods of estrus detection in mice are either imprecise, or time consuming and stressful for the subjects. The authors evaluated the efficiency of a novel method of estrus detection, based on electrical resistance of vaginal mucus and compared it to the gold standard method of vaginal cytology, and other commonly used methods in the lab. Moreover, authors assessed animal behavior stress responses to the different tests. Overall, the manuscript is scientifically sound and is well written and the appropriate references are cited. The data presented are conclusive and convincing and show the benefits of adopting this tool for estrus detection. Authors can improve the quality of the manuscript by addressing the comments below:   

There is some text written in red font.

Lines 70-96: I’m not sure you need to be this thorough in explaining the estrous cycle for your manuscript. You don’t need to edit this as it is accurate, but perhaps simplifying this would make it more attractive for the reader. Just a suggestion.

Lines 101 – 134: Consider Combining into one paragraph.

Methods:

I suggest you describe the overall experimental timeline before the methodology for the different estrus detection methods. For example ”30 pubertal mice were used in the experiment. All animals were assessed daily for stage of the estrous cycle by assessing vaginal opening status, vaginal cytology and MED, etc” The way it is currently written, I’m not sure if you had a group of mice which you assessed estrous stage by vaginal opening only, and another group where you assessed estrous stage by vaginal cytology, etc, or if you performed all assessments in parallel in all individuals. An image summarizing the timeline with experimental procedures and sample collections would be greatly beneficial for this manuscript and would be clearer for the reader.

2.2. and 2.3. You provide a good explanation on how the assessments are done. However, you do not explain why and when such measurements were taken. 

Line 178: What is Rs?

Lines 305-310: This is methodology, not results. It is fine to refresh the reader of this here, but this information is missing in the methodology. See comment above.

Lines 340-342: This is methodology, not results. It is fine to refresh the reader of this here, but this information is missing in the methodology.

Lines 426-428: This is methodology, not results. It is fine to refresh the reader of this here, but this information is missing in the methodology.

Lines 319-320: I know you mentioned that comparisons were made between vaginal opening and vaginal smear analysis, but please clarify that the vaginal smear analysis was used as gold standard for these comparisons.

Lines 404 – 406: I suggest simplifying this phrase to “intraperitoneal injections of gonadotropin agonists, initiating the maturation of numerous follicles”. Please use “oocyte” instead of “egg”.

Line 418: Compared (you’re missing a “d”)

Italicize “P in your P – values

Lines 486-494: Have you considered calculating sensitivity and specificity of this test?

Line 521: Estrus stage of the cycle

Author Response

Comments 1:

There is some text written in red font.

Lines 70-96: I’m not sure you need to be this thorough in explaining the estrous cycle for your manuscript. You don’t need to edit this as it is accurate, but perhaps simplifying this would make it more attractive for the reader. Just a suggestion.

Response 1:

Corrected

Comments 2:

Lines 101 – 134: Consider Combining into one paragraph.

Response 2:

Corrected

Comments 3:

Methods:

I suggest you describe the overall experimental timeline before the methodology for the different estrus detection methods. For example ”30 pubertal mice were used in the experiment. All animals were assessed daily for stage of the estrous cycle by assessing vaginal opening status, vaginal cytology and MED, etc” The way it is currently written, I’m not sure if you had a group of mice which you assessed estrous stage by vaginal opening only, and another group where you assessed estrous stage by vaginal cytology, etc, or if you performed all assessments in parallel in all individuals. An image summarizing the timeline with experimental procedures and sample collections would be greatly beneficial for this manuscript and would be clearer for the reader.

Response 3:

We added in Materials and Methods: Thirty pubertal female mice (ten for each line: CD-1, FVB/N, and C57Bl6/J) were assessed daily for stage of the estrous cycle by assessing vaginal opening status, MEDProTM measurements, and vaginal smear cytology.

Comments 4:

2.2. and 2.3. You provide a good explanation on how the assessments are done. However, you do not explain why and when such measurements were taken.

Response 4:

We added in Materials and Methods, see above.

Comments 5:

Line 178: What is Rs?

Response 5:

RS is an current resistance of the epithelial cell layer of vaginal mucosa. Line 176-182

Comments 6:

Lines 305-310: This is methodology, not results. It is fine to refresh the reader of this here, but this information is missing in the methodology. See comment above.

Response 6:

We added in Materials and Methods, see above.

Comments 7:

Lines 340-342: This is methodology, not results. It is fine to refresh the reader of this here, but this information is missing in the methodology.

Response 7:

We added in Materials and Methods, see above.

Comments 8:

Lines 426-428: This is methodology, not results. It is fine to refresh the reader of this here, but this information is missing in the methodology.

Response 8:

Corrected

Comments 9:

Lines 319-320: I know you mentioned that comparisons were made between vaginal opening and vaginal smear analysis, but please clarify that the vaginal smear analysis was used as gold standard for these comparisons.

Response 9:

Corrected

Comments 10:

Lines 404 – 406: I suggest simplifying this phrase to “intraperitoneal injections of gonadotropin agonists, initiating the maturation of numerous follicles”.

Response 10:

Not corrected. We suggest to keep it non-modified.

Comments 11:

Please use “oocyte” instead of “egg”.

Response 11:

Corrected

Comments 12:

Line 418: Compared (you’re missing a “d”)

Response 12:

Corrected

Comments 13:

Italicize “P in your P – values

Response 13:

Corrected

Comments 14:

Lines 486-494: Have you considered calculating sensitivity and specificity of this test?

Response 14:

Our data revealed that values equal or greater than the threshold of 6 kOm in 76%, 64% and 76% for CD-1, FVB/N and C57Bl6/j mouse strains, respectively. These values were not detected for diestrus cycles in any of the investigated strains.

Comments 15:

Line 521: Estrus stage of the cycle

Response 15:

Corrected

Reviewer 2 Report

Comments and Suggestions for Authors

Comments about the manuscript:

“Advancing 3Rs: The Mouse Estrus Detector (MED) as a Low-Stress, Painless, and Efficient Tool for Estrus Determination in Mice”

Determining the stages of the estrous cycle in mice is essential for its multiple uses as a model animal, whether wild or genetically modified. The most precise possible detection of the estrous cycle including the reproductive phases is relevant in the ethical context of the 3Rs – Replacement, Reduction and Refinement. The manuscript presented here concerns results obtained independently by two laboratories on the efficiency and accuracy of the Elmi Ltd. Mouse Estrus Detector (MED). To do this, the authors analyzed the effectiveness of the detector in five strains of female mice (CD1, FVB/N, C57Bl6/J, B6D2F1 and Swiss). The results showed the qualities of the MEDProTM detector and the importance of using modern and efficient methods to meet the requirements of ethical conditions.

This study, which concerns the effectiveness of a new device making it possible to detect the stages of the estrous cycle of several strains of mice, provides interesting and useful results. In my opinion, however, the manuscript requires improvements before considering its publication. Here are some remarks

Page 1, line 38. Use italics to write “Mus musculus”.

Page 3, line 140, Materials and methods. “In this study, a number of experiments were carried out”: This part should give more details. What were the experiences? The descriptions of the different groups of animals used are not very clear. A table with a summary of the experiments, groups of mice, method used will be very useful to the reader. This part needs to be developed.

Page 4, line 200. “miniature probe”: specify the size of the probe.

Page 5: table 1 is not referred to in the text.

Page 6, figure 2. This figure is not referred to in the text. It could be grouped with Figure 1 which would be devoted to the device tested.

Page 7, figure 3. Add a scale bar to each micrograph.

Page 8, lines 306-307. “we conducted a comprehensive series of experiments involving CD-1, FVB/N, and C57Bl6/J female mice (n=10 for each strain)”: A description of these experiences is missing. See note about Materials and Methods.

Page 8, lines 311-312. “We collected a total of 330 vaginal smear samples from females of the investigated strains.”: Specify how many smears were taken per female per female.

Discussion. In the discussion, a comparison of the effectiveness of the MEDProTM system with other methods of detecting the estrous cycle would have seemed relevant to me.

Author Response

Comments 1:

Page 1, line 38. Use italics to write “Mus musculus”.

Response 1:

Corrected

Comments 2:

Page 3, line 140, Materials and methods. “In this study, a number of experiments were carried out”: This part should give more details. What were the experiences? The descriptions of the different groups of animals used are not very clear. A table with a summary of the experiments, groups of mice, method used will be very useful to the reader. This part needs to be developed.

Response 2:

We have added: see detailed description below. In Materials and methods below detailly described: 2.2 Vaginal opening observation, 2.3 General Principle of MEDProTM Measurement and 2.4 Vaginal smear cytology for mouse estrous cycle stage identification. All 3 groups of animals: CD-1, FVB/N and C57Bl6/J are detailly represented in Supplementary Table 1. Each animal from all 3 groups presented in this table has data on assessments of vaginal opening, AR measurements, and vaginal smear samples cytology (daily evaluations).

Comments 3:

Page 4, line 200. “miniature probe”: specify the size of the probe.

Response 3:

Corrected

Comments 4:

Page 5: table 1 is not referred to in the text.

Response 4:

Corrected

Comments 5:

Page 6, figure 2. This figure is not referred to in the text. It could be grouped with Figure 1 which would be devoted to the device tested.

Response 5:

Corrected

Comments 6:

Page 7, figure 3. Add a scale bar to each micrograph.

Response 6:

The experiments, including vaginal smear were performed in 2019, and pictures were saved in electronic format. The scale bar was not added, because it was not important for the estrous cycle stage identification.  

Comments 7:

Page 8, lines 306-307. “we conducted a comprehensive series of experiments involving CD-1, FVB/N, and C57Bl6/J female mice (n=10 for each strain)”: A description of these experiences is missing. See note about Materials and Methods.

Response 7:

Please, see note above, the description of these experiments presented in Materials and Methods.

Comments 8:

Page 8, lines 311-312. “We collected a total of 330 vaginal smear samples from females of the investigated strains.”: Specify how many smears were taken per female per female.

Response 8:

Corrected

Comments 9:

Discussion. In the discussion, a comparison of the effectiveness of the MEDProTM system with other methods of detecting the estrous cycle would have seemed relevant to me.

Response 9:

Added
